# Emergent competition shapes top-down versus bottom-up control in multi-trophic ecosystems

Zhijie Feng[1]*, Robert Marsland III[1¤], Jason W. Rocks[1], Pankaj Mehta[1,2,3]*

**1** Department of Physics, Boston University, Boston, Massachusetts, United States of America, **2** Biological Design Center, Boston University, Boston, Massachusetts, United States of America, **3** Faculty of Computing and Data Science, Boston University, Boston, Massachusetts, United States of America

¤ Current address: Pontifical University of the Holy Cross, Rome, Italy

\* zjfeng@bu.edu (ZF); pankajm@bu.edu (PM)

**Data Availability Statement:** All code associated with the manuscript is available at https://github.com/Emergent-Behaviors-in-Biology/Multi-trophic-ecosystem.

## Abstract

Ecosystems are commonly organized into trophic levels—organisms that occupy the same level in a food chain (e.g., plants, herbivores, carnivores). A fundamental question in theoretical ecology is how the interplay between trophic structure, diversity, and competition shapes the properties of ecosystems. To address this problem, we analyze a generalized Consumer Resource Model with three trophic levels using the zero-temperature cavity method and numerical simulations. We derive the corresponding mean-field cavity equations and show that intra-trophic diversity gives rise to an effective "emergent competition" term between species within a trophic level due to feedbacks mediated by other trophic levels. This emergent competition gives rise to a crossover from a regime of top-down control (populations are limited by predators) to a regime of bottom-up control (populations are limited by primary producers) and is captured by a simple order parameter related to the ratio of surviving species in different trophic levels. We show that our theoretical results agree with empirical observations, suggesting that the theoretical approach outlined here can be used to understand complex ecosystems with multiple trophic levels.

## Author summary

Ecosystems are commonly organized into trophic levels—organisms that occupy the same level in a food chain (e.g., plants, herbivores, carnivores). In this study, we use methods originating from spin glass physics to theoretically analyze the statistical properties of large and diverse ecosystems with multiple trophic levels. Our analysis successfully quantifies the strength of intra-level competitions and results in a simple criteria for determining whether an ecosystem exhibits top-down control (e.g. herbivores populations are limited by predators) or bottom-up control (e.g. herbivores populations are limited by the availability of plants). Somewhat surprisingly, we find that whether a system exhibits top-

**Funding:** This work was supported by NIH NIGMS grant 1R35GM119461 award to PM. The funders had no role in study design, data collection and analysis, decision to publish, or preparation of the manuscript.

**Competing interests:** The authors have no competing interests.

down or bottom-up control dependent solely on the ratio of surviving species at different trophic levels.

## 1 Introduction

A defining feature of natural ecosystems is their immense complexity. This complexity is especially prominent in diverse ecosystems with many different types of interacting species and resources. It is common to think about ecosystems in terms of energy flows: energy is harvested from the environment by primary producers (e.g., photosynthetic organisms) and then flows through the ecosystem via the food chain [1]. Energy flows in ecosystems can be understood by organizing species into trophic levels: sets of organisms that occupy the same level in a food chain [2, 3]. A classic example is a food pyramid consisting of three trophic levels: primary producers (organisms that can directly harvest energy from the environment, e.g., plants), primary consumers (organisms that derive energy by consuming the primary producers, e.g., herbivores), and secondary consumers (organisms that derive energy from predation of the primary consumers, e.g., carnivores).

Understanding the ecological consequences of such trophic structures remains an open problem in modern ecology [4]. To simplify the complexity of such systems, previous theoretical studies have often ignored the effects of intra-trophic level diversity, focusing entirely on coarse-grained energy flows between trophic levels. This approach has yielded numerous insights, including the incorporation of top-down and bottom-up control, the role of vertical diversity, and scaling laws for organism size and metabolism under different regimes [5–9]. However, the use of coarse-grained trophic levels makes it difficult to understand the effects of species diversity and competition on ecosystem structure and function. Given the importance of biodiversity and competition as ecological drivers [10–12], there is a need for theoretical approaches that allow for the simultaneous study of trophic structure, diversity, and competition.

Here, we address this shortcoming by building upon a series of recent works that utilize ideas from statistical physics to understand the effects of competition and diversity in large ecosystems with many species [13–23]. In particular, we focus on a three trophic level generalization of the MacArthur Consumer Resource Model (MCRM), a prominent ecological model for competition. First introduced by Levins and MacArthur, the MCRM considers an ecosystem with two trophic levels corresponding to primary producers (resources) and primary consumers [24–26]. In the MCRM, consumers are defined by a set of consumer preferences that encode how likely each consumer is to consume each resource. Competition occurs when species have similar consumer preferences and hence occupy similar niches [27].

Our model generalizes the MCRM in two ways. First, we introduce an additional trophic level into the system. In addition to the primary producers, or resources, of the bottom level and consumers of the top level, we introduce a middle level where species play the role of both consumers and resources. Second, inspired by the success of "random ecosystems" in capturing the properties of real ecosystems [20, 21, 23, 28, 29], we consider a large ecosystem with many species at each trophic level where all consumer preferences and ecological parameters are drawn from random distributions. The use of random parameters has a long history in theoretical ecology and allows us to model typical behaviors we expect to encounter [30].

To study this model, we make use of analytic calculations based on the zero-temperature cavity method and numerical simulations. In particular, we derive analytic expressions for steady-state distributions of species at all three trophic levels, allowing us to explore the

interplay between trophic structure, diversity, and competition and construct ecological phase diagrams for ecosystem behaviors.

## 2 Result

### 2.1 Multi-trophic consumer resource model

**2.1.1 Theoretical setup.** We begin by presenting a generalization of the MCRM to multi-trophic systems. We consider an ecosystem consisting of three trophic levels: a bottom trophic level consisting of $M_R$ species of primary producers (e.g., plants) whose abundances we denote by $R_P$ ($P = 1, \ldots, M_R$), a middle trophic level consisting of $M_N$ species of primary consumers (e.g., herbivores) with abundances $N_i$ ($i = 1, \ldots, M_N$), and a top level consisting of $M_X$ secondary consumers (e.g. carnivores) $X_\alpha$ ($\alpha = 1, \ldots, M_X$). We note that while we present results for three levels, this model and the corresponding mean-field cavity solutions presented in the next section can easily be generalized to an arbitrary number of trophic levels (see S1 Text).

The dynamics of the ecosystem are described by a set of non-linear differential equations of the form

$$
\begin{aligned}
\frac{dX_\alpha}{dt} &= X_\alpha \left[ \eta_X \sum_j d_{\alpha j} N_j - u_\alpha \right] \\
\frac{dN_i}{dt} &= N_i \left[ \eta_N \sum_Q c_{iQ} R_Q - m_i - \sum_\beta d_{\beta i} X_\beta \right] \\
\frac{dR_P}{dt} &= R_P \left[ K_P - R_P - \sum_j c_{jP} N_j \right],
\end{aligned}
\tag{1}
$$

where $c_{iQ}$ is a $M_N \times M_R$ matrix of consumer preferences for the the $M_N$ primary consumers and $d_{\beta i}$ is a $M_X \times M_N$ matrix of consumer preferences for the $M_X$ secondary consumers, and $\eta_X, \eta_N \in [0, 1]$ account for finite efficiency of biomass conversion. When $\eta_X$ and $\eta_N$ are close to one, energy is very efficiently transferred across trophic levels. In contrast, when these values are close to zero, energy cannot be efficiently harvested from lower trophic levels.

We also define the carrying capacity $K_P$ for each primary producer $P$, along with the death rates $m_i$ for each primary consumer $i$ and $u_\alpha$ for each secondary consumer. These dynamics share key assumptions with the original MCRM on how energy flows from the environment to different species and how species interact with each other. The major difference between the two models is the addition of the intermediate trophic level, $N_i$, where species act as both "resources" to the secondary consumers above and "consumers" of the primary producers below. To provide intuition, we will use the terms "carnivores", "herbivores" and "plants" in later text to refer to "secondary consumers", "primary consumers" and "primary producers," respectively.

In Fig 1(a), we depict an example of this model graphically with species organized into three distinct trophic levels composed of carnivores, herbivores, and plants. At the bottom, there is a constant flux of energy into the system from the environment. In the absence of herbivores, plants in the bottom level grow logistically to their carrying-capacity $K_P$. Predation reduces the resource abundances at the bottom, resulting in an upward flow of energy. Energy returns to the environment through death, represented by death rates $u_\alpha$ and $m_i$.

In addition to energy flows, the ecosystem is structured by competition between species through the consumer preference matrices $d_{\alpha j}$ and $c_{iP}$. As in the original MCRM, species within a trophic level with similar consumer preferences compete more and consequently, can

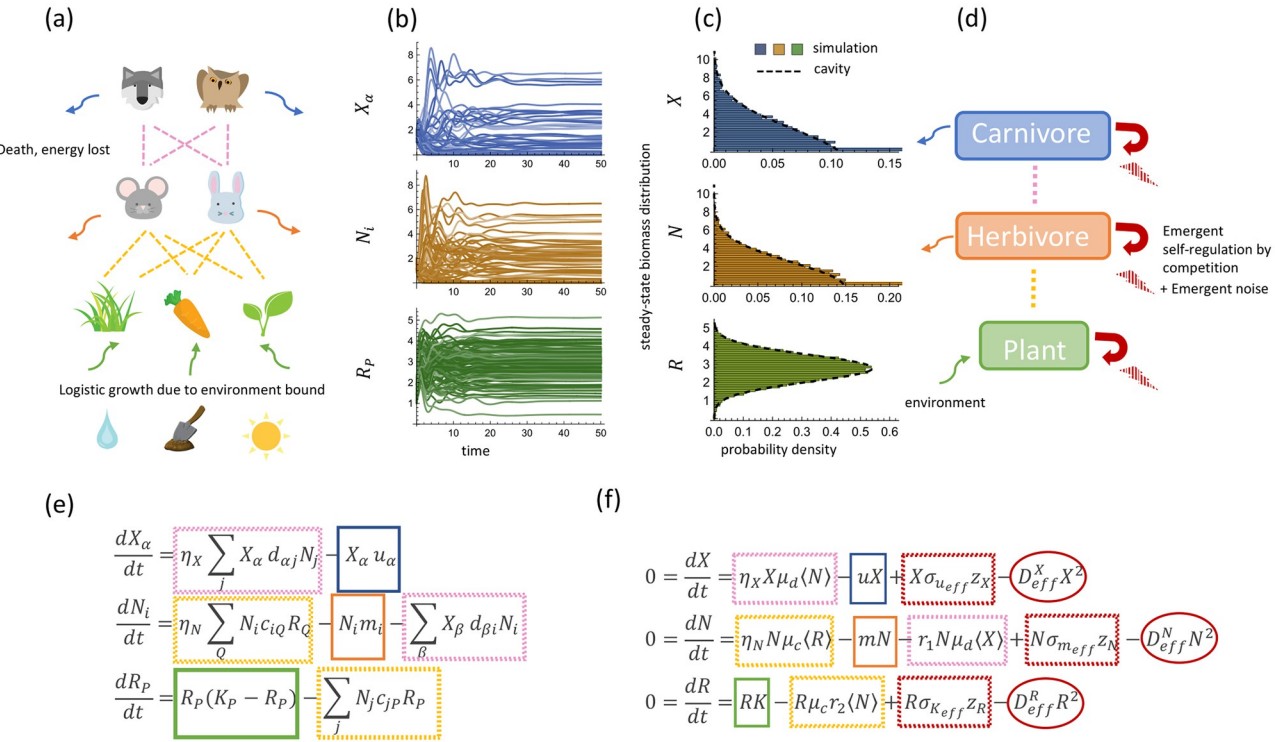

**Fig 1. (a)** Schematic of food web interaction with three-level trophic structure, colored corresponding to the model equation in (e). **(b)** Simulated dynamics of a system with $M_X = 50$ species of carnivores, $M_N = 56$ herbivores and $M_R = 62$ plants with $k = 4$, $m = 1$, $u = 1$, $\sigma_c = \sigma_d = 0.5$, $\mu_c = \mu_d = 1$, $\eta_X = 0.8$, $\eta_N = 0.6$, $\sigma_k = \sigma_m = \sigma_u = 0.1$. **(c)** Histograms of the steady-state distributions reached by simulated dynamics of 200 systems of the same condition in (b) and the distributions predicted by our cavity solutions. Note that the peaks at zero correspond to delta functions for extinct species. For visual clarity, they are not shown to full scale, but also agree with the cavity predictions (see S4 Fig). **(d)** Schematic of the coarse-grained view of the three-level trophic structure, colored corresponding to equations in (f). **(e)** Equations of the three-level trophic structure model corresponding to (a). **(f)** Effective mean-field (TAP) equations for steady-states have additional emergent competition and random variation terms proportional to $D^{\mathcal{A}}_{eff}$ ($\mathcal{A} = X, N, R$) and $\sigma_{\mathcal{B}}$ ($\mathcal{B} = u_{eff}, m_{eff}, K_{eff}$), respectively. In 1(a), the icon of the wolf is adapted from "Creative-Tail-Animal-wolf" by Creative Tail licensed under CC BY 4.0, and all other icons are adapted from cliparts from Openclipart licensed under CC0 1.0 DEED.

competitively exclude each other [24]. One qualitatively new feature of the multi-trophic MCRM is that niches in the herbivore level are defined by both the consumer preferences $c_{iP}$ for the species in the bottom level and the ability to avoid predation by carnivores through their consumer preferences $d_{\alpha j}$. The consumer preferences $c_{iP}$ and $d_{\alpha j}$ control both energy flows between trophic levels and competition between species within a trophic level.

To proceed, we specify the free parameters $c_{jP}$, $d_{\alpha j}$, $K_P$, $m_i$, and $u_\alpha$. Because we are interested in the *typical* behaviors of large multi-trophic ecosystems (the thermodynamic limit, $M_R$, $M_N$, $M_X \gg 1$), we follow a rich tradition in theoretical ecology and statistical physics of drawing parameters randomly from distributions [30, 31]. We consider the case where the consumer preferences $d_{ia}$ are drawn independently and identically with mean $\mu_d/M_N$ and standard deviation $\sigma_d/\sqrt{M_N}$. We parameterize the variation in $d_{ia}$ in terms of the random variables $\gamma_{id}$ so that

$$d_{\alpha i} = \frac{\mu_d}{M_N} + \sigma_d \gamma_{\alpha i}$$

$$\langle \gamma_{\alpha i} \rangle = 0, \qquad \langle \gamma_{\alpha i} \gamma_{\beta j} \rangle = \frac{\delta_{\alpha \beta} \delta_{ij}}{M_N}.$$

(2)

Similarly, we draw the consumer preferences $c_{iA}$ independently and identically with mean $\mu_c/M_R$ and standard deviation $\sigma_c/\sqrt{M_R}$, parameterized in terms of the random variables $\epsilon_{jP}$,

$$c_{iP} = \frac{\mu_c}{M_R} + \sigma_c \epsilon_{iP}$$

$$\langle \epsilon_{iP} \rangle = 0, \qquad \langle \epsilon_{iP} \epsilon_{jQ} \rangle = \frac{\delta_{ij}\delta_{PQ}}{M_R}.$$

(3)

For convenience, we choose to scale the means and variances of the consumer preferences with the number of species, $1/M_N$ or $1/M_R$. We note that this does not affect the generality of our results, but greatly simplifies the mathematical treatment in the thermodynamic limit.

With the knowledge that niches overlaps of consumers depend on the ratio of the mean versus standard deviation of consumer preferences [26], we fix $\mu_c = 1$ and $\mu_d = 1$. In most simulations we also choose to draw the consumer preferences from Gaussian distributions. However, we note that our results also generalize to other distributions that obey the above statistical properties such as the uniform distribution where coefficients are strictly positive (see S5 Fig).

Finally, we choose the parameters $u_\alpha$, $m_i$, and $K_P$ to be independent Gaussian random variables with means $u$, $m$, and $k$ and standard deviations $\sigma_u$, $\sigma_m$, and $\sigma_K$, respectively. We also fix $\eta_N = \eta_X = 1$, $\sigma_K = 0.1$, $\sigma_u = 0.1$, and $\sigma_m = 0.1$.

In Fig 1(b), we depict the typical dynamical evolution of such a system, where the biomass of each species fluctuates for a finite time before reaching equilibrium. While the dynamics of consumer-resource models can display rich behavior, for instance chaos and limit cycles when consumer-resource interactions are asymmetric (non-reciprocal) [32], we choose to focus on the case where the interactions are symmetric. In the physical regime where the mean values of each parameter and the initial biomass of each species is positive, we have found that our numerical simulations always converge to a unique globally stable steady-state for the ecological dynamics. While we currently lack a rigorous proof, we suspect that this reflects the fact that the the multi-trophic consumer resource model possesses a Lyapunov function similar to the two trophic layer system [33, 34].

**2.1.2 Derivation of cavity solutions.**   In a very large ecosystem, understanding the detailed behaviors of each species is not possible. For this reason, we focus on developing a statistical description of the ecological dynamics in steady-state. This is made possible by the observation that the each species interacts with many other species in the ecosystem, allowing us to characterize the effects of interactions using a mean-field theory. This philosophy originates from the statistical physics of spin glasses and has more recently been imported into the study of ecological systems [15, 26, 35–37].

To derive the mean-field cavity equations for the steady-state behavior, we focus on the thermodynamic limit, $M_R, M_N, M_X \to \infty$, while holding the ratios of species fixed, $r_1 = M_X/M_N$ and $r_2 = M_N/M_R$. The key idea of the zero-temperature cavity method is to relate properties of an ecosystem of size $(M_X, M_N, M_R)$ to an ecosystem with size $(M_X + 1, M_N + 1, M_R + 1)$ where a new species is added at each trophic level. For large ecosystems, the effects of the new species are small enough to capture with perturbation theory, allowing us to derive self-consistent equations. On a technical level, we assume that our ecosystem is self-averaging and replica symmetric [31].

Under these assumptions, we find that "typical" species at each trophic level, represented by the random variables $X$, $N$, and $R$, follow truncated Gaussian distributions, given by

$$
\begin{aligned}
X &= \max\left[0, \frac{g_{eff}^X + \sigma_{g_{eff}^X} z_x}{D_{eff}^X}\right] \\[2ex]
N &= \max\left[0, \frac{g_{eff}^N + \sigma_{g_{eff}^N} z_N}{D_{eff}^N}\right] \\[2ex]
R &= \max\left[0, \frac{g_{eff}^R + \sigma_{g_{eff}^R} z_R}{D_{eff}^R}\right],
\end{aligned}
\tag{4}
$$

where $z_X$, $z_N$, $z_R$ are independent Gaussian random variables with zero mean and unit variance and the effective parameters are given by the expressions

$$
\begin{aligned}
g_{eff}^X &= -u + \eta_X \mu_d \langle N \rangle \\[1ex]
g_{eff}^N &= -m - r_1 \mu_d \langle X \rangle + \eta_N \mu_c \langle R \rangle \\[1ex]
g_{eff}^R &= K - \mu_c r_2 \langle N \rangle \\[1ex]
\sigma_{g_{eff}^X}^2 &= \eta_X^2 \sigma_d^2 \langle N^2 \rangle + \sigma_u^2 \\[1ex]
\sigma_{g_{eff}^N}^2 &= \eta_N^2 \sigma_c^2 \langle R^2 \rangle + \sigma_d^2 r_1 \langle X^2 \rangle + \sigma_m^2 \\[1ex]
\sigma_{g_{eff}^R}^2 &= \sigma_k^2 + \sigma_c^2 r_2 \langle N^2 \rangle \\[1ex]
D_{eff}^X &= -\eta_X \sigma_d^2 \nu \\[1ex]
D_{eff}^N &= \eta_N \sigma_c^2 \kappa - \eta_X r_1 \sigma_d^2 \chi \\[1ex]
D_{eff}^R &= 1 - \eta_N r_2 \sigma_c^2 \nu.
\end{aligned}
\tag{5}
$$

We use the notation $\langle . \rangle$ to denote averages over the distributions in Eq (4). With this notation, we define the the mean abundance of species at each trophic level, $\langle R \rangle$, $\langle N \rangle$, and $\langle X \rangle$, the second moments of the species abundances, $\langle R^2 \rangle$, $\langle N^2 \rangle$, and $\langle X^2 \rangle$, and the mean susceptibility of each trophic level biomass with respect to the change of direct energy flow in or out from the environment at that level, $\chi = \langle \frac{\partial X}{\partial u} \rangle$, $\nu = \langle \frac{\partial N}{\partial m} \rangle$, and $\kappa = \langle \frac{\partial R}{\partial K} \rangle$.

In S1 Text, we provide a detailed explanation of how Eqs (4) and (5) can be used to derive a set of self-consistent cavity equations to solve for the means and second moments of the abundances, the susceptibilities, and the fraction of surviving species at each trophic level. Fig 1(c) shows a comparison between the predictions of the steady-state distributions of $R$, $N$, and $X$ and direct numerical simulation of Eq (1). We can see that there is remarkable agreement with simulations results. This suggests that the cavity method accurately captures the large scale properties of multi-trophic ecosystems.

Our calculations are exact in the thermodynamic limit where there are infinite number of species in the regional pool at each trophic level. To understand the finite size corrections, we performed numerical simulations for ecosystems of different sizes. As can be seen in S4 Fig, our analytic predictions agree well with numerics even for modest size ecosystems with of order 20 species.

## 2.2 Emergent competition

**2.2.1 Effective coarse-grained picture.**   The cavity solutions from the previous section allow us to calculate the biomass of species in each trophic level. A key feature of these equations is that the effect of species competition is summarized by self-consistent Thouless-Anderson-Palmer (TAP) corrections proportional to the parameters $D_{eff}^X$, $D_{eff}^R$, and $D_{eff}^N$ [see Eq (4)]. We now show that these three parameters have a natural interpretations as encoding the "emergent competition" between species within each trophic level mediated by interactions with other trophic levels.

To see this, we note that Eq (4) can also be rearranged to give effective steady-state equations for a typical species at each level,

$$0 = \frac{dX}{dt} = X\left[g_{eff}^X + \sigma_{g_{eff}^X} z_x - D_{eff}^X X\right]$$

$$0 = \frac{dN}{dt} = N\left[g_{eff}^N + \sigma_{g_{eff}^N} z_N - D_{eff}^N N\right] \tag{6}$$

$$0 = \frac{dR}{dt} = R\left[g_{eff}^R + \sigma_{g_{eff}^R} z_R - D_{eff}^R R\right].$$

We emphasize that these equations are only valid at steady-state and capturing the actual coarse-grained dynamics would involve solving for the full dynamical mean-field theory equations. However, rewriting the steady-state solutions in this form clarifies the meaning of $D_{eff}^X$, $D_{eff}^N$, and $D_{eff}^R$. Species at each trophic level have an effective description in terms of a logistic growth equation, with the parameters $D_{eff}^X$, $D_{eff}^N$, and $D_{eff}^R$ controlling how much individuals within each trophic level compete with each other. In addition, Eq (6) demonstrates that the species within each trophic level can be thought of as having effective carrying capacities drawn from Gaussian distributions with means $g_{eff}^X$, $g_{eff}^N$, and $g_{eff}^R$, and standard deviations $\sigma_{g_{eff}^X}$, $\sigma_{g_{eff}^N}$, and $\sigma_{g_{eff}^R}$, respectively. This coarse-grained view of the resulting ecological dynamics is illustrated in Fig 1(d) with the correspondence between terms in the original and coarse-grained steady state equations depicted in Fig 1(e) and 1(f).

**2.2.2 Relation to species packing.**   To better understand the origins of this emergent competition, we relate $D_{eff}^X$, $D_{eff}^N$, and $D_{eff}^R$ to the number of surviving species and the species packing fractions. One of the key results of niche theory is the competition-exclusion principle which states that the number of species that can be packed into an ecosystem is bounded by the number of realized (available) niches [24, 38]. In Consumer Resource Models (CRMs), the number of realized niches is set by the number of surviving species at each trophic level. For the top trophic level, the competitive exclusion principle states that the number of surviving carnivores $M_X^*$ must be smaller than the number of surviving herbivores $M_N^*$,

$$M_X^* \leq M_N^*. \tag{7}$$

For herbivores which reside in the middle trophic levels, niches are defined by both the ability to consume plants in the bottom trophic level and the ability to avoid predation by carnivores in the top trophic level. For this reason, competitive exclusion on herbivores takes the form

$$M_N^* \leq M_R^* + M_X^*, \tag{8}$$

where $M_R^*$ is the number of plants that survive at steady-states. In other words, for herbivores there are $M_R^* + M_X^*$ potential realized niches of which $M_N^*$ are filled.

The cavity equations derived from Eq (4) naturally relate species packing fractions to the effective competition coefficients $D_{eff}^X$, $D_{eff}^N$, and $D_{eff}^R$ in Eq (5). Before proceeding, it is helpful to define the ratio

$$f = \frac{M_R^* + M_X^* - M_N^*}{M_N^* - M_X^*}$$

$$= \frac{\text{\#of unfilled realized niches in middle trophic level}}{\text{\#of unfilled realized niches in top trophic level}}$$

(9)

and the ratio $\phi_N = M_N^*/M_N$, the fraction of species in the regional species pool that survive in the middle level. Using these ratio, in the S1 Text, we show that the effective competition coefficients can be written

$$D_{eff}^X = \frac{\eta_X \sigma_d^2}{\eta_N \sigma_c^2 r_2} \frac{1}{f}$$

$$D_{eff}^N = \eta_N \phi_N \sigma_c^2 r_2 f$$

(10)

$$D_{eff}^R = 1 + \frac{1}{f}.$$

These expressions show that there is a direct relationship between the amount of emergent competition at each trophic level and the number of occupied niches (species packing properties). The effective competition coefficient for herbivores, $D_{eff}^N$, decreases with the number of unoccupied niches in the top trophic level, and shows a non-monotonic dependence on the number of species in the middle level. Moreover, direct examination of the expressions in Eq (10) shows that the amount of competition in the top and bottom levels is positively correlated, in agreement with the well-established ecological intuition for trophic levels separated by an odd number of levels [39–41].

To better understand these expressions, we used the cavity equations to numerically explore how emergent competition parameters at each trophic level depend on the diversity of the regional species pool (as measured by $\sigma_c^2$ and $\sigma_d^2$) and environmental parameters ($k$, $u$, $r_1$, and $r_2$). We summarize these results in Table 1 and S2 and S3 Figs. One consistent prediction of our model is that the effective competition in each level always decreases with the size of the regional species pool of that level. This effect has been previously discussed in the ecological literature under the names "sampling effect" and "variance in edibility" [39, 42–44]. We also find that in almost all cases, the effective competition coefficients change monotonically as model

**Table 1. Effect of changing model parameters on emergent competition and the relative strength of top-down versus bottom-up control.** The last column refers to related hypothesis or observations summarized in Table 2. The symbols indicate increase ↑, or decrease ↓. The table is also valid with symbols flipped (↑ replaced by ↓, and vice versa). See S2 and S3 Figs for corresponding numerical simulations.

| Label | Change to ecosystem ↑ | Parameter change | $D_{eff}^X$ | $D_{eff}^N$ | $D_{eff}^R$ | $\frac{D_{eff}^{N.top}}{D_{eff}^{N.top}+D_{eff}^{N.bottom}}$ |
|---|---|---|---|---|---|---|
| 1 | carnivore species richness | $r_1 \uparrow$ | ↓ | ↑ | ↓ | ↑ |
| 2 | herbivore species richness | $r_1 \downarrow\ r_2 \uparrow$ | ↑ | ↓ | ↑ | ↓ |
| 3 | plant species richness | $r_2 \downarrow$ | mostly ↓ | ↑ | ↓ | slightly ↑ or ↓ |
| 4 | carnivore preference variance | $\sigma_d \uparrow$ | ↑ | mostly ↑ | mostly ↓ | mostly ↑ |
| 5 | herbivore preference variance | $\sigma_c \uparrow$ | ↓ | ↑ | mostly ↑ | ↓ |
| 6 | death rate of carnivore | $u \uparrow$ | ↑ | ↓ | ↑ | ↓ |
| 7 | energy influx to plant | $k \uparrow$ | ↑, or ↓, or ↑ then ↓ | ↑, or ↓ then ↑ | ↓, or ↑ then ↓ | ↑ |

parameters are varied. One notable exception to this is the effect of changing the amount of energy supplied to the ecosystem as measured by the average carrying capacity $k$ of plants (resources) in the bottom level. We find that often the amount of emergent competition in the bottom level, $D_{eff}^R$, first increases with $k$ then decreases, and this non-monotonic behavior propagates to $D_{eff}^N$ and $D_{eff}^X$. Finally, we observe the that $D_{eff}^X$, $D_{eff}^N$, and $D_{eff}^R$ generally increase with $\sigma_c$ and $\sigma_d$.

## 2.3 Order parameters for top-down vs bottom-up control

Ecosystems are often robust to certain classes of perturbations while being fragile to others. For instance, ocean ecosystems are known to react much more drastically to loss of nutrients and sunlight than loss of big predator fishes [45]. Motivated by observations such as these, ecologists often classify ecosystems into two broad categories depending on the type of perturbations they are most sensitive to: ecosystems with bottom-up control and ecosystems with top-down control [Fig 2(a) and 2(b), respectively]. Bottom-up control describes ecosystems that are susceptible to perturbations of the bottom trophic level, while top-down control describes ecosystems that are susceptible to perturbation of the top trophic level. In S1 Text, we present a simple toy-model that illustrates these concepts.

For example, Fig 2(c) shows simulations from an ecosystem that exhibits bottom-up control. Changing the average carrying capacity $k$ of plants in the bottom level increases the biomass of herbivores and predators at higher trophic levels. In contrast, the middle and bottom

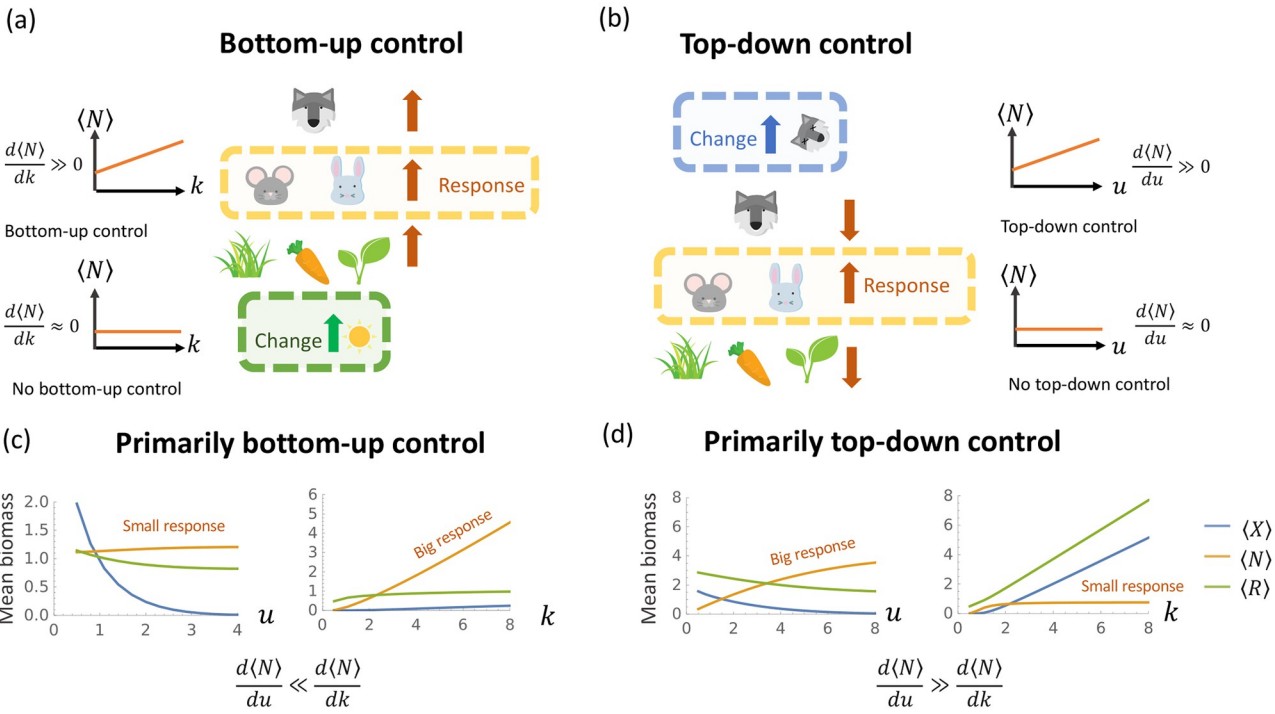

**Fig 2. (a)** Bottom-up control. Increasing the total energy energy influx $k$ to primary producers in the bottom trophic level increases the average biomass $\langle N \rangle$ of herbivores in the middle trophic level. **(b)** Top-down control. Increasing the death rate $u$ of predators in the top trophic level increases the biomass of the middle trophic level. **(c)** Average biomass at each trophic level obtained from cavity solutions as a function of $u$ with $k = 1$, $r_1 = 0.2$, $r_2 = 1.2$, $\sigma_c = 0.5$, $\sigma_d = 0.5$, (left) and as a function of $k$ with $u = 3$ (right). **(d)** Same as (c) except $r_1 = 1.3$, $r_2 = 0.3$. This difference in parameters is an example of the herbivore species richness increase described in Table I. In 2(a)-(b), the icon of the wolf is adapted from "Creative-Tail-Animal-wolf" by Creative Tail licensed under CC BY 4.0, and all other icons are adapted from cliparts from Openclipart licenced under CC0 1.0 DEED.

trophic levels are relatively insensitive to changes in the average death rate $u$ of predators in the top trophic level. Fig 2(d) shows a simulation of an ecosystem that exhibits top-down control. Increasing the death rate of predators results in increased populations of herbivores (middle level) but decreased populations of predators (top level) and plants (bottom levels). This alternating behavior across trophic levels is characteristic of ecosystems with top-down control. In contrast, the biomass in the middle is largely insensitive to changes in the carrying capacity $k$ of plants in the bottom level.

**2.3.1 Measuring top-down versus bottom-up control.**   Historically, it was assumed that ecosystems could not simultaneously exhibit both top-down and bottom-up control [40, 46, 47]. However, recent evidence—such as the impact of overfishing on aquatic ecosystems—has overturned this view leading to a consensus that most ecosystems are impacted by both types of control and that their relative importance can shift over time [8, 48–51]. Building on these ideas, recent theoretical works suggest that ecosystems can shift between bottom-up and top-down control dominated regimes as one varies model parameters [9, 52–54]. Here, we revisit and extend these works using CRMs and our cavity solution to investigate the effects of species diversity and other environmental factors on top-down versus bottom-up control.

One important challenge we must overcome is the lack of a consensus in the ecology literature on how to quantify bottom-up versus top-down control in an ecosystem. Empirical studies often use the structure of correlations in time series of species abundances across trophic levels [48, 49, 51]. An alternative experimental approach is based on the ability to create small ecosystems with slightly different environments and/or compositions of predators in the top trophic level [43, 53, 55]. Unfortunately, conclusions between these two frameworks often do not agree with each other [56]. For this reason, it is necessary to revisit the problem of quantifying bottom-up and top-down control.

One common proposal for characterizing the response of ecosystems to perturbations in both empirical and theoretical studies is looking at the biomass distribution of different trophic levels. It has been argued that in a system with bottom-up control, we should expect the total biomass of the bottom trophic level to be larger than the total biomass of the top trophic level, $M_R\langle R\rangle > M_X\langle X\rangle$ In contrast, in a system with top-down control, we expect the opposite, $M_X\langle X\rangle > M_R\langle R\rangle$. Other existing theoretical works make use of derivatives to measure the results of various perturbations [8]. The most direct quantities we can look at are the derivatives $\frac{d\langle N\rangle}{dk}$ and $\frac{d\langle N\rangle}{du}$ that capture the change in the average biomass $\langle N\rangle$ of species in the middle trophic level in response to changes in the average carrying capacity $k$ of plants (bottom trophic level) and changes in the average death rate $u$ of carnivores (top trophic level). An intuitive measurement to compare these two quantities may be the ratio $\frac{d\langle N\rangle}{dk} / \frac{d\langle N\rangle}{du} \in [0, \infty)$, but converting it to fraction $\frac{d\langle N\rangle}{dk} / \left(\frac{d\langle N\rangle}{du} + \frac{d\langle N\rangle}{dk}\right) \in [0, 1]$ gives us a more well-behaved order parameter.

**2.3.2 Cavity-inspired order parameters.**   Here, we use our cavity solution to the multi-trophic MCRM to propose two informative and intuitive order parameters to assess whether an ecosystem has top-down or bottom-up control. We then show that they qualitatively agree with each other and the definitions based on derivatives discussed above (see Fig 3).

**Biomass-based order parameter**. To create our first order parameter, we rewrite the form of the effective growth rate for the biomass in the middle trophic level [Eq (5)] as

$$g_{eff}^N = -m + g_{eff}^{N,top} + g_{eff}^{N,bottom}$$

$$g_{eff}^{N,top} = -r_1\mu_d\langle X\rangle, \qquad g_{eff}^{N,bottom} = \eta_N\mu_c\langle R\rangle$$

(11)

**Fig 3.** **(a)** The emergent competition coefficient for the middle level, $D_{eff}^N$, can be written as the sum of two terms resulting from feedbacks from the top trophic level, $D_{top}^N$, and the bottom trophic level, $D_{bottom}^N$. The order parameter $D_{top}^N/(D_{top}^N + D_{bottom}^N)$ quantifies the sensitivity to top-down versus bottom-up control. **(b)** Comparison of three order parameters discussed in the main text for measuring top-down versus bottom-up control: $\frac{d\langle N\rangle}{dk} / \left(\frac{d\langle N\rangle}{dk}\frac{d\langle N\rangle}{du}\right)$, $g_{eff}^{N,top}/(g_{eff}^{N,top} + g_{eff}^{N,bottom})$, $D_{eff}^{N,top}/(D_{eff}^{N,top} + D_{eff}^{N,top})$. Each point corresponds to an ecosystem with different choices of $k$, $u$, $r_1$, and $r_2$. In 3(a), the icon of the wolf is adapted from "Creative-Tail-Animal-wolf" by Creative Tail licensed under CC BY 4.0, and all other icons are adapted from cliparts from Openclipart licensed under CC0 1.0 DEED.

Each of the three terms in $g_{eff}^N$ captures distinct ecological processes of herbivores in the middle level: (i) the first term proportional to $m$ is the intrinsic death rate, (ii) the middle term, $g_{eff}^{N,top}$, captures the effect of predation due to carnivores in the top trophic level, and (iii) the third term, $g_{eff}^{N,bottom}$, measures the consumption of plants in the bottom trophic level. Based on this interpretation, we propose the following ratio as a natural measure of top-down versus bottom-up control:

$$\left| \frac{g_{eff}^{N,top}}{g_{eff}^{N,top} + g_{eff}^{N,bottom}} \right| = r_1 \frac{\mu_d\langle X\rangle}{\mu_d\langle X\rangle + \eta_N\mu_c\langle R\rangle} . \tag{12}$$

This ratio measures the relative contributions of the top and bottom trophic levels on the growth rate of species in the middle level. Notice that in addition to the biomass, this definition also accounts for the strength of competition between species via $\mu_c$ and $\mu_d$, along with differences in the regional species pool sizes via the extra factor $r_1 = M_X/M_N$.

**Species packing-based order parameter**. We also construct an order parameter for top-down versus bottom-up control based on the relative contributions of the top and bottom trophic levels to the emergent competition coefficient of the middle level, $D_{eff}^N$. Using the

definition in Eq (5), we rewrite this coefficient as

$$D_{eff}^N = D_{eff}^{N,top} + D_{eff}^{N,bottom}$$

$$D_{eff}^{N,top} = \eta_X r_1 \sigma_d^2 \chi, \qquad D_{eff}^{N,bottom} = \eta_N \sigma_c^2 \kappa \tag{13}$$

where $D_{eff}^{N,top}$ and $D_{eff}^{N,bottom}$ capture feedbacks from the top and bottom trophic levels, respectively, onto the middle level. Based on this, we define the corresponding order parameter as

$$\frac{D_{eff}^{N,top}}{D_{eff}^{N,top} + D_{eff}^{N,bottom}} = \frac{-\eta_X r_1 \sigma_d^2 \chi}{\eta_N \sigma_c^2 \kappa - \eta_X r_1 \sigma_d^2 \chi} = \frac{M_X^*}{M_N^*}, \tag{14}$$

where in the second equality we have used the cavity solutions to relate the susceptibilities to species packing fractions (see S1 Text). Since competition exclusion leads to $M_X^* < M_N^*$, this order parameter corresponds to simply the fraction of realized niches that are filled in the top level. By construction, if $D_{eff}^{N,top}/D_{eff}^{N,top} + D_{eff}^{N,bottom} > 0.5$, then an ecosystem exhibits more top-down control than bottom-up control, while $D_{eff}^{N,top}/D_{eff}^{N,top} + D_{eff}^{N,bottom} < 0.5$ indicates the opposite is true.

Finally, we note that somewhat surprisingly this order parameter does not *explicitly* depend on the biomass conversion efficiencies $\eta_X$ and $\eta_N$. For this reason, within our model, the effect of imperfect energy conversion manifests itself only through species packing fractions.

**2.3.3 Order parameters are consistent with ecological intuitions.** To better understand if these species-packing order parameters capture traditional intuitions about top-down versus bottom-up control, we compare $D_{eff}^{N,top}/(D_{eff}^{N,top} + D_{eff}^{N,bottom})$, $|g_{eff}^{N,top}/(g_{eff}^{N,top} + g_{eff}^{N,bottom})|$, and $\frac{d\langle N\rangle}{dk}/\left(\frac{d\langle N\rangle}{dk} + \frac{d\langle N\rangle}{du}\right)$ to each other for ecosystems where we varied the model parameters $k$, $u$, $r_1$, and $r_2$. The results are shown in Fig 3. Notice that all three quantities are highly correlated, especially at the two extreme ends. This suggests that the order parameter $D_{eff}^{N,top}/(D_{eff}^{N,top} + D_{eff}^{N,bottom})$ is an especially useful tool to infer whether an ecosystems is more susceptible to bottom-up or top-down control, as it requires us to simply count the number of surviving species in the top and middle trophic levels. If we have more occupied niches in the top level than unoccupied niches, $M_X^*/M_N^* > 0.5$, the ecosystem is more susceptible top-down control. If the opposite is true $M_X^*/M_N^* < 0.5$, then the ecosystem is more susceptible to bottom-up control.

## 2.4 Phase diagram changes with diversity

Having established that $D_{eff}^{N,top}/(D_{eff}^{N,top} + D_{eff}^{N,bottom})$ is a good order parameter for assessing the relative importance of bottom-up and top-down control, we now use this quantity to construct phase diagrams. One important ecological parameter of interest is the total energy entering the ecosystem. In our model, this is controlled by the average carrying capacity $k$ of plants at the bottom trophic level. Another ecologically important parameter is the predator death rate $u$ which controls the biomass in the top trophic level. The number and diversity of species in the ecosystem is set by $r_1 = M_X/M_N$ and $r_2 = M_N/M_R$, which determine the relative sizes of the regional species pools at each trophic level, and $\sigma_c$ and $\sigma_d$, which control the trait diversity via the standard deviation of consumer preferences. Fig 4(a) shows the dependence of $D_{eff}^{N,top}/(D_{eff}^{N,top} + D_{eff}^{N,bottom})$ on $k$, $u$, $r_1$, and $r_2$, while the phase diagrams in Fig 4(b) explore the dependence of $D_{eff}^{N,top}/(D_{eff}^{N,top} + D_{eff}^{N,bottom})$ on $\sigma_c$ and $\sigma_d$.

Notice that $D_{eff}^{N,top}/(D_{eff}^{N,top} + D_{eff}^{N,bottom})$ always increases with $k$ and decreases with $u$. These trends agrees with our expectation that ecosystems are more likely to exhibit top-down

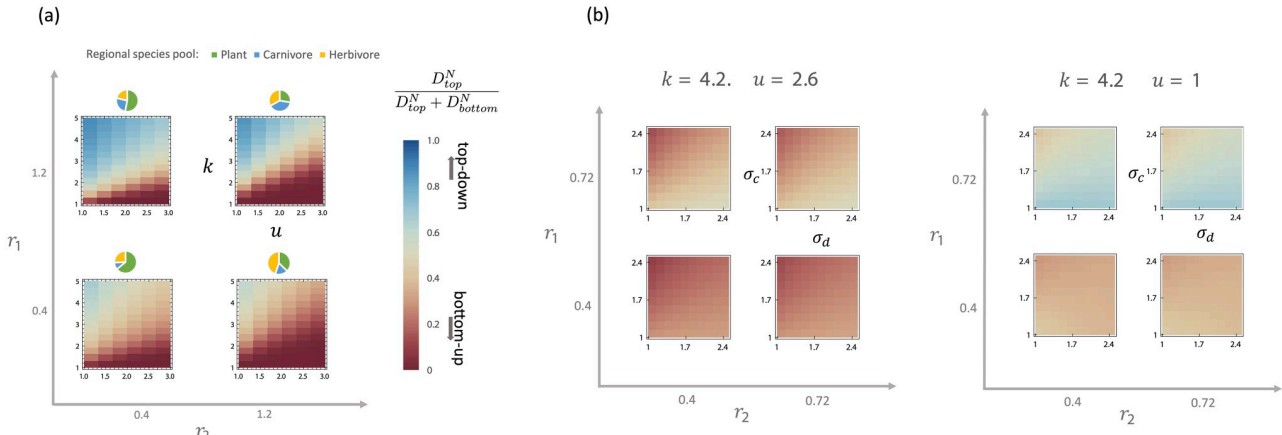

**Fig 4. (a)** as a function of energy influx to primary producers $k$ and death rate of carnivores $u$ for four different ratios of regional species pool, $r_1, r_2 \in \{0.4, 1.2\}$, indicated by the pie chart, with $\sigma_c = 0.5$ and $\sigma_d = 0.5$, and **(b)** as a function of the species trait diversity, $\sigma_d$ and $\sigma_c$, for four different ratios of regional species pools with environmental parameters $k = 4.2$, $u = 2.6$ and $k = 4.2$, $u = 1$.

(bottom-up) control when they are limited by the top (bottom) trophic level. A larger $k$ reduces the survival stress on species in the middle level from food limitations, decreasing the importance of bottom-up control. Analogously, a larger $u$ reduces the stress from predators, decreasing the importance of top-down control.

The amount of top-down control $D_{eff}^{N,top}/(D_{eff}^{N,top} + D_{eff}^{N,bottom})$ also increases with $r_1$ and decreases with $r_2$. This observation is consistent with what is known in the ecological literature as the "sampling effect", where larger regional species pool size leads to a higher fitness of surviving species [42, 44]. A smaller $r_1$ and larger $r_2$ correspond to increasing the size of the regional species pool of the middle trophic level relative to the top level or bottom level, respectively. This increases the odds that herbivores can cope with the survival stress from predators and/or more efficiently consume plants.

In Fig 4(b), we also show how $D_{eff}^{N,top}/(D_{eff}^{N,top} + D_{eff}^{N,bottom})$ depends on the trait diversity via $\sigma_c$ and $\sigma_d$. Notice that the amount of top-down control decreases as the diversity of the herbivores increases via $\sigma_c$, while it increases as predators in the top trophic level become more diverse via $\sigma_d$. One notable exception is a small region in phase space with large $k$, small $u$, small $r_1$, and small $\sigma_c$, where $D_{eff}^{N,top}/(D_{eff}^{N,top} + D_{eff}^{N,bottom})$ decreases with $\sigma_d$. A similar dependence on $\sigma_d$ is observed for $D_{eff}^{R}$, suggesting that this idiosyncratic behavior may be mediated by a complex feedback involving both carnivores and plants.

## 2.5 Predictions, proposed experiments, and comparison to ecological literature

Tables 1 and 2 compare the prediction of our model for emergent ecosystem properties to the ecological literature. We primarily focus on predictions concerning how the effective competition strength at each trophic level ($D_{eff}^{X}, D_{eff}^{N}, D_{eff}^{R}$) and the relative strength of top-down versus bottom-up control ($D_{eff}^{N,top}/(D_{eff}^{N,top} + D_{eff}^{N,bottom})$) vary with the number of species ($r_1, r_2$), species diversity ($\sigma_c, \sigma_d$) and environmental parameters ($k, u$). In particular, Table 2 summarizes the predictions from our model in simple terms and presents observations/hypothesis from the ecological literature consistent with our model predictions. Overall, it is quite striking how

**Table 2. Existing observations and hypothesis in trophic ecology that relates to the model behavior in Fig 4 and S2 and S3 Figs.** The first column refers to the related model behavior summarized in Table 1.

| Model behavior | Observation/hypothesis | References |
|---|---|---|
| 1 | Increased species richeness in a trophic level lead to higher biomass and resource comsumption in its level | [42, 57, 58] |
| 2, 5 | Herbivore diversity may increase bottom-up control and decrease top-down control through complementarity | [45, 56] |
| 2, 3 | Increasing prey richness increase the chance of resistance to predator (variance in edibility hypothesis) | [43] |
| 1, 3 | Ecosystem are much more sensitive to loss of predators diversity than plants diversity | [55, 59] |
| 4, 5 | Increasing consumer generalism (horizontal niche breadth) reduces or alters the impact of consumer richness on prey biomass | [60, 61] |
| 7 | Increasing the resources to a system can be destabilizing (paradox of enrichment). | [62] |
| 7 | Bottom-up cascade: An increase in primary producer will be passed on to the predators in a three-level food chain. | [63] |
| 3 | Increased plant diversity results in reduced herbivory | [64] |
| 1 | Increased predator diversity results in reduced herbivory | [65] |
| 6 | Removing top predators by hunting, fishing and whaling has lead to flourishing mesopredators | [66] |
| 1, 2, 6 | Top-down cascade: Removal of predators from a food chain with odd number of levels reduces plant biomass, vice versa for even number | [40, 41] |
| 7 | Bottom-up effect becomes weaker when nutrient is abundant | [48, 67] |

many different qualitative observations/hypothesis are reproduced by our generalized MCRM with three trophic levels.

The predictions of our models can also be directly tested using current experimental techniques. One prediction of our theory is that whether a three trophic level ecosystem exhibits top-down or bottom-up control can be determined by counting the number of species in the middle and top trophic levels. In principle, this can be done using perturbative experiments on synthetic microcosms under different conditions [54]. Another interesting direction for testing our predictions is to use existing food web data, focusing on the number of coexisting species and biomass at each trophic level. One potential setting for doing this is to compare properties of aquatic and terrestrial food webs since aquatic ecosystems are generically thought to be more susceptible to top-down control than terrestrial ecosystems [68, 69]. It may be also interesting to fit the model to actual ecosystems, for instance by making use of rank-abundance curves data for real systems to fix the biomass distributions of the model.

## 3 Method

### 3.1 Cavity derivations

**3.1.1 Three-level consumer resource model.**   The derivation in this section follows closely with the derivation in Ref. [26]. To derive the cavity solutions for the steady-state behavior of our three-level Consumer Resource Model, we focus on the limit $M_R, M_N, M_X \to \infty$, while holding the ratios of species $r_1 = M_X/M_N$ and $r_2 = M_N/M_R$ fixed. The key idea of the cavity method is to relate properties of an ecosystem of size $(M_X, M_N, M_R)$ to an ecosystem with size $(M_X + 1, M_N + 1, M_R + 1)$ where a new species has been added at each trophic level, while keeping all other parameters the same. We start by evaluating Eq (1) at steady-state and adding one

new species at each level represented by an index of 0,

$$
\begin{aligned}
0 \quad &= \frac{\mathrm{d}X_\alpha}{\mathrm{d}t} = X_\alpha \left[ \mu_d \langle N \rangle + \eta_X \sigma_d \sum_j \gamma_{\alpha j} N_j + \eta_X \sigma_d \gamma_{\alpha 0} N_0 - u_\alpha \right] \\
0 \quad &= \frac{\mathrm{d}N_i}{\mathrm{d}t} = \\
& N_i \left[ \mu_c \langle R \rangle + \eta_N \sigma_c \sum_Q \epsilon_{iQ} R_Q + \eta_N \sigma_c \epsilon_{i0} R_0 - m_i - \mu_d r_1 \langle X \rangle - \sigma_d \sum_\beta \gamma_{\beta i} X_\beta - \sigma_d \gamma_{0i} X_0 \right] \\
0 \quad &= \frac{\mathrm{d}R_P}{\mathrm{d}t} = R_P \left[ K_P - R_P - \mu_c r_2 \langle N \rangle - \sigma_c \sum_j \epsilon_{jP} N_j - \sigma_c \epsilon_{0P} N_0 \right].
\end{aligned}
\tag{15}
$$

We have also substituted the decomposition of the consumer preferences into their mean and varying parts according to Eqs (2) and (3). Note that all sums in this derivation are assumed to start at index 1 unless otherwise specified. Furthemore, we have defined the mean species abundances

$$
\langle X \rangle = \frac{1}{M_X} \sum_{\beta=0}^{M_X} X_\beta, \qquad \langle N \rangle = \frac{1}{M_N} \sum_{j=0}^{M_N} N_j, , \qquad \langle R \rangle = \frac{1}{M_R} \sum_{Q=0}^{M_R} R_Q.
\tag{16}
$$

We also introduce a new steady-state equation for each of the new species,

$$
\begin{aligned}
0 \quad &= \frac{\mathrm{d}X_0}{\mathrm{d}t} = X_0 \left[ \mu_d \langle N \rangle + \eta_X \sigma_d \sum_j \gamma_{0j} N_j + \eta_X \sigma_d \gamma_{00} N_0 - u_0 \right] \\
0 \quad &= \frac{\mathrm{d}N_0}{\mathrm{d}t} = \\
& N_0 \left[ \mu_c \langle R \rangle + \eta_N \sigma_c \sum_Q \epsilon_{0Q} R_Q + \eta_N \sigma_c \epsilon_{00} R_0 - m_0 - \mu_d r_1 \langle X \rangle - \sigma_d \sum_\beta \gamma_{\beta 0} X_\beta - \sigma_d \gamma_{00} X_0 \right] \\
0 \quad &= \frac{\mathrm{d}R_0}{\mathrm{d}t} = R_0 \left[ K_0 - R_0 - \mu_c r_2 \langle N \rangle - \sigma_c \sum_j \epsilon_{j0} N_j - \sigma_c \epsilon_{00} N_0 \right].
\end{aligned}
\tag{17}
$$

Next, we interpret the additional terms added to Eq (15) as perturbations to the growth/death rate parameters,

$$
\begin{aligned}
\delta u_\alpha \quad &= -\eta_X \sigma_d \gamma_{\alpha 0} N_0 \\
\delta m_i \quad &= -\eta_N \sigma_c \epsilon_{i0} R_0 + \sigma_d \gamma_{0i} X_0 \\
\delta K_P \quad &= -\sigma_c \epsilon_{0P} N_0.
\end{aligned}
\tag{18}
$$

Using these perturbations, we can write down the new steady-states in terms of a Taylor expansion of the original ones without the new species. Using Einstein summation notation

for repeated index (summing from index 1 instead of 0), these equations take the form

$$
\begin{aligned}
X_\alpha &= X_{\alpha/0} + \chi_{\alpha\beta}\delta\mu_\beta + v_{\alpha i}\delta m_i + \kappa_{\alpha P}\delta K_P, \\
N_i &= N_{i/0} + \chi_{i\alpha}\delta\mu_\alpha + v_{ij}\delta m_j + \kappa_{iP}\delta K_P, \\
R_P &= R_{P/0} + \chi_{P\alpha}\delta\mu_\alpha + v_{Pi}\delta m_i + \kappa_{PQ}\delta K_Q.
\end{aligned}
\tag{19}
$$

where we have defined the susceptibility matrices

$$
\begin{aligned}
\chi_{\alpha\beta} &= \frac{\partial X_\alpha}{\partial u_\beta} & v_{\alpha j} &= \frac{\partial X_\alpha}{\partial m_j} & \kappa_{\alpha Q} &= \frac{\partial X_\alpha}{\partial K_Q} \\
\chi_{i\beta} &= \frac{\partial N_i}{\partial u_\beta} & v_{ij} &= \frac{\partial N_i}{\partial m_j} & \kappa_{iQ} &= \frac{\partial N_i}{\partial K_Q} \\
\chi_{P\beta} &= \frac{\partial R_P}{\partial u_\beta} & v_{Pj} &= \frac{\partial R_P}{\partial m_j} & \kappa_{PQ} &= \frac{\partial R_P}{\partial K_Q}.
\end{aligned}
\tag{20}
$$

Now we focus on the equations for the 0-th species in each level. We substitute the expanded form of the new-steady states into Eq (17) and only keep the lowest order terms in large $M_X$, $M_N$, and $M_R$. It is straightforward to show via the the central limit theorem that each of the sums can be approximated in terms of a mean and variance component (see Ref. [26] for more details). Performing these approximations, we find the following self-consistency equations for the abundances of the new species:

$$
\begin{aligned}
0 &= X_0[g_{eff}^X + \sigma_{g_{eff}^X} z_X - D_{eff}^X X_0] \\
0 &= N_0[g_{eff}^N + \sigma_{g_{eff}^N} z_N - D_{eff}^N N_0] \\
0 &= R_0[g_{eff}^R + \sigma_{g_{eff}^R} z_R - D_{eff}^R R_0].
\end{aligned}
\tag{21}
$$

where $z_X$, $z_N$, $z_R$ are Gaussian variables with zero mean and unit variance and we have defined

$$
\begin{aligned}
g_{eff}^X &= -u + \eta_X \mu_d \langle N\rangle \\
g_{eff}^N &= -m - r_1 \mu_d \langle X\rangle + \eta_N \mu_c \langle R\rangle \\
g_{eff}^R &= K - \mu_c r_2 \langle N\rangle \\
\sigma_{g_{eff}^X}^2 &= \eta_X^2 \sigma_d^2 \langle N^2\rangle + \sigma_u^2 \\
\sigma_{g_{eff}^N}^2 &= \eta_N^2 \sigma_c^2 \langle R^2\rangle + \sigma_d^2 r_1 \langle X^2\rangle + \sigma_m^2 \\
\sigma_{g_{eff}^R}^2 &= \sigma_k^2 + \sigma_c^2 r_2 \langle N^2\rangle \\
D_{eff}^X &= -\eta_X \sigma_d^2 v \\
D_{eff}^N &= \eta_N \sigma_c^2 \kappa - \eta_X r_1 \sigma_d^2 \chi \\
D_{eff}^R &= 1 - \eta_N r_2 \sigma_c^2 v
\end{aligned}
\tag{22}
$$

with

$$\langle X^2 \rangle = \frac{1}{M_X} \sum_{\beta} X_{\beta}^2, \quad \langle N^2 \rangle = \frac{1}{M_N} \sum_{j} N_j^2,,, \quad \langle R^2 \rangle = \frac{1}{M_R} \sum_{Q} R_Q^2,$$

$$\chi = \frac{1}{M_X} \sum_{\beta} \frac{\partial X_{\beta}}{\partial u_{\beta}}, \qquad v = \frac{1}{M_N} \sum_{j} \frac{\partial N_j}{\partial m_j},,, \qquad \kappa = \frac{1}{M_R} \sum_{Q} \frac{\partial R_Q}{\partial K_Q}. \tag{23}$$

Rearranging the self-consistency equations above, we find that $X_0$, $N_0$, and $R_0$ follow truncated Gaussian distributions of the form

$$X_0 \quad = \max \left[ 0, \frac{g_{eff}^X + \sigma_{g_{eff}^X} z_X}{D_{eff}^X} \right)$$

$$N_0 \quad = \max \left[ 0, \frac{g_{eff}^N + \sigma_{g_{eff}^N} z_N}{D_{eff}^N} \right) \tag{24}$$

$$R_0 \quad = \max \left[ 0, \frac{g_{eff}^R + \sigma_{g_{eff}^R} z_R}{D_{eff}^R} \right).$$

Finally, we make use of the fact that here is nothing special about species 0, i.e., the system is "self-averaging" so the biomass distribution of one species over many systems is the same as that of many species in one system. We compute the averages $\langle X \rangle$, $\langle N \rangle$, $\langle R \rangle$, $\langle X^2 \rangle$, $\langle N^2 \rangle$, $\langle R^2 \rangle$, $\chi$, $v$, and $\kappa$ by simply taking appropriate averages of $X_0$, $N_0$. and $R_0$. This gives us our final set of self-consistency equations,

$$\langle X \rangle \quad = \frac{\sigma_{g_{eff}^X}}{D_{eff}^X} w_1(\Delta_{g_{eff}^X}) \tag{25}$$

$$\langle N \rangle \quad = \frac{\sigma_{m_{eff}}}{D_{eff}^N} w_1(\Delta_{g_{eff}^N}) \tag{26}$$

$$\langle R \rangle \quad = \frac{\sigma_{K_{eff}}}{D_{eff}^R} w_1(\Delta_{g_{eff}^R}) \tag{27}$$

$$\langle X^2 \rangle \quad = \left( \frac{\sigma_{g_{eff}^X}}{D_{eff}^X} \right)^2 w_2(\Delta_{g_{eff}^X}) \tag{28}$$

$$\langle N^2 \rangle \quad = \left( \frac{\sigma_{m_{eff}}}{D_{eff}^N} \right)^2 w_2(\Delta_{g_{eff}^N}) \tag{29}$$

$$\langle R^2 \rangle \quad = \left( \frac{\sigma_{K_{eff}}}{D_{eff}^R} \right)^2 w_2(\Delta_{g_{eff}^R}) \tag{30}$$

$$\chi \quad = \left\langle \frac{\partial X}{\partial u} \right\rangle = \frac{w_0(\Delta_{g_{eff}^X})}{D_{eff}^X} \tag{31}$$

$$v \quad = \left\langle \frac{\partial N}{\partial m} \right\rangle = -\frac{w_0(\Delta_{g_{eff}^N})}{D_{eff}^N} \tag{32}$$

$$\kappa \quad = \left\langle \frac{\partial R}{\partial K} \right\rangle = \frac{w_0(\Delta_{g_{eff}^R})}{D_{eff}^R} \tag{33}$$

where

$$\Delta_{g_{eff}^R} = \frac{g_{eff}^R}{\sigma_{g_{eff}^R}}, \qquad \Delta_{g_{eff}^X} = \frac{g_{eff}^X}{\sigma_{g_{eff}^X}}$$

$$\Delta_{g_{eff}^N} = \frac{g_{eff}^N}{\sigma_{g_{eff}^N}} \tag{34}$$

and we define the integrals

$$w_n(\Delta) = \int_{-\Delta}^{\infty} \frac{dx}{\sqrt{2\pi}} (x + \Delta)^n e^{-x^2/2}. \tag{35}$$

Then the $j^{th}$ moment of truncated Gaussian $y = \max\left(0, \frac{a+cz}{b}\right)$, with positive $c/b$, can be written in terms of the integral

$$\langle y^j \rangle = \left(\frac{c}{b}\right)^j \int_{-a/c}^{\infty} \frac{dz}{\sqrt{2\pi}} e^{-\frac{z^2}{2}} \left(z + \frac{a}{c}\right)^j = \left(\frac{c}{b}\right)^j w_j\left(\frac{a}{c}\right).$$

**3.1.2 *N*-level consumer resource model.**   Here we present our generalized consumer resource model an arbitrary number of levels and relaxed assumptions on intra-species competition and biomass conversion efficiency. We consider $N$ levels with $M^i$ species on each level ($i = 1, \ldots, N$). We use $i = 1$ to represent the bottom level and $i = N$ for the top level. The abundance $B_\mu^i$ for the $\mu^{th}$ follows the dynamics

$$\frac{1}{B_\mu^i} \frac{dB_\mu^i}{dt} = g_\mu^i - D^i B_\mu^i + \sum_v \eta^i \alpha_{\mu v}^{i,i-1} B_v^{i-1} - \sum_v \alpha_{v\mu}^{i+1,i} B_v^{i+1} \tag{36}$$

where $\alpha^{i,\,i-1}$ is the consumer preference matrix of species on the level $i$ feeding on species on level $i-1$ beneath them. The top and bottom levels have boundary conditions $\alpha^{1,0} = \alpha^{N,\,N-1} = 0$. In general, any variable with superscript $i < 1$ or $i > N$ are 0. In analogy to the three-level model, we consider random consumer preference matrices with mean and variance

$$\alpha_{\mu v}^{i,i-1} = \frac{\mu_\alpha^i}{M^i} + \sigma_\alpha^i d_{\mu v}^i$$

$$\langle d_{\mu v}^{i,i-1} \rangle = 0, \qquad \langle d_{\mu v}^{i,i-1} d_{\alpha\beta}^{i,i-1} \rangle = \sigma_\alpha^i \frac{\delta_{\mu\alpha}\delta_{v\beta}}{M^i} \tag{37}$$

where we have parameterized the variation in terms of the random variables $d_{\alpha\beta}^{i,i-1}$. We also define the growth/death rates $g_\mu^i$ to be independent with mean $g^i$ and standard deviation $\sigma_g^i$ for each level. We note that in physical ecosystems $g_i$ should be positive for level $i = 1$ and negative for higher levels. Finally, we define the new parameters $D^i$ to account for intra-species

competition in level $i$ that is not mediated by consumption of or predation by other levels, and $\eta^i \in [0, 1]$ to account for finite efficiency of biomass conversion. Previously in the main text, we assumed $D^3$, $D^2 = 0$ for carnivores and herbivores and $D^1 = 1$ for plants, and all energy conversion are perfect $\eta^2 = \eta^3 = 1$.

The following self-consistency cavity equation can be obtained by generalizing the previous results from the three-level model. For the $i^{th}$ level, the abundance $B_i$ follows a truncated Gaussian distribution

$$B^i = \max\left(0, \frac{g^i_{eff} + \sigma_{g^i_{eff}} z^i}{D^i_{eff}}\right) \tag{38}$$

where $z^i$ are zero-mean Gaussian random variables with unit variance and effective variables

$$
\begin{aligned}
g^i_{eff} &= g^i - r^{i+1}\mu^{i+1}_\alpha \langle B^{i+1}\rangle + \eta^i \mu^{i-1}_\alpha \langle B^{i-1}\rangle \\
\sigma_{g^i_{eff}} &= \sqrt{(\eta^i)^2 (\sigma^{i-1}_\alpha)^2 q^{i-1} + (\sigma^{i+1}_\alpha)^2 r^{i+1} q_{i+1} + (\sigma^i_g)^2} \\
D^i_{eff} &= D^i + \eta^i (\sigma^{i-1}_\alpha)^2 \chi^{i-1} + \eta^{i+1} r^{i+1} (\sigma^{i+1}_\alpha)^2 \chi^{i+1}
\end{aligned}
\tag{39}
$$

with definitions

$$
\begin{aligned}
r^{i+1} &= M^{i+1}/M^i, \qquad \langle B^0\rangle = \langle B^{N+1}\rangle = 0. \\
q^i &= \langle (B^i)^2\rangle, \qquad \chi^i = \left\langle \frac{\partial B^i}{\partial g^i}\right\rangle
\end{aligned}
\tag{40}
$$

We derive the self-consistency equations by taking appropriate averages. For $N$ levels, there are a total of $3N$ equations,

$$
\begin{aligned}
\langle B^i\rangle &= \frac{\sigma_{g^i_{eff}}}{D^i_{eff}} w_1(\Delta_{g^i_{eff}}), \\
\langle (B^i)^2\rangle &= \left(\frac{\sigma_{g^i_{eff}}}{D^i_{eff}}\right)^2 w_2(\Delta_{g^i_{eff}}), \\
\chi^i &= \frac{w_0(\Delta_{g^i_{eff}})}{D^i_{eff}}.
\end{aligned}
\tag{41}
$$

where $\Delta_{g^i_{eff}} = g^i_{eff}/\sigma_{g^i_{eff}}$. We can define top-down to bottom-up ratio for each level

$$\frac{D^{i,top}_{eff}}{D^{i,top}_{eff} + D^{i,bottom}_{eff}} = \frac{\eta^{i+1} r^{i+1} (\sigma^{i+1}_\alpha)^2 \chi^{i+1}}{\eta^{i+1} r^{i+1} (\sigma^{i+1}_\alpha)^2 \chi^{i+1} + \eta^i (\sigma^{i-1}_\alpha)^2 \chi^{i-1}} \tag{42}$$

Moreover, we can again write down the effective mean-field (TAP) equations for steady states with effective competition. Defining the effective competition coefficients $D^i_{eff}$, we can derive coarse-grained equations for each layer at steady-state,

$$0 = \frac{dB^i}{dt} = B^i[g^i - D^i_{eff}B^i - r^{i+1}\mu^{i+1}_\alpha \langle B^{i+1}\rangle + \eta^i \mu^{i-1}_\alpha \langle B^{i-1}\rangle + \sigma^i_B z^i], \tag{43}$$

which look very similar to the coarse-grained model of multi-level food chains in [9], except that the noise and competition are emergent.

**Effective competition coefficients.**   Using the cavity equation, we can solve explicitly for the three susceptibilities in Eqs (31)–(33),

$$
\begin{aligned}
\chi &= -\frac{\eta_N \sigma_c^2 w_0(\Delta_{g_{eff}^X})(w_0(\Delta_{g_{eff}^R}) - r_2 w_0(\Delta_{g_{eff}^N}) + r_1 r_2 w_0(\Delta_{g_{eff}^X}))}{\eta_X \sigma_d^2 (w_0(\Delta_{g_{eff}^N}) - r_1 w_0(\Delta_{g_{eff}^X}))} \\[2mm]
v &= -\frac{w_0(\Delta_{g_{eff}^N}) - r_1 w_0(\Delta_{g_{eff}^X})}{\eta_N \sigma_c^2 (w_0(\Delta_{g_{eff}^R}) - r_2 w_0(\Delta_{g_{eff}^N}) + r_1 r_2 w_0(\Delta_{g_{eff}^X}))} \\[2mm]
\kappa &= w_0(\Delta_{g_{eff}^R}) - r_2 w_0(\Delta_{g_{eff}^N}) + r_1 r_2 w_0(\Delta_{g_{eff}^X}).
\end{aligned}
\tag{44}
$$

Next, we observe that the integrals for the zeroth moments measure the fraction of species in the regional species pool that survives at steady state, i.e.,

$$
\begin{aligned}
w_0(\Delta_{g_{eff}^X}) &= M_X^*/M_X, \\[2mm]
w_0(\Delta_{g_{eff}^N}) &= M_N^*/M_N, \\[2mm]
w_0(\Delta_{g_{eff}^R}) &= M_R^*/M_R.
\end{aligned}
\tag{45}
$$

Also, recall the definitions

$$
\begin{aligned}
r_1 &= M_X/M_N, \\[2mm]
r_2 &= M_N/M_R.
\end{aligned}
\tag{46}
$$

Substituting these expressions into Eq (44), the susceptibilities become

$$
\begin{aligned}
\chi &= -\frac{\eta_N \sigma_c^2}{\eta_X \sigma_d^2} \frac{M_X^*}{M_X} \frac{M_N}{M_R} \frac{M_R^* - M_N^* + M_X^*}{M_N^* - M_X^*}, \\[2mm]
v &= -\frac{M_R}{\eta_N M_N \sigma_c^2} \frac{M_N^* - M_X^*}{M_R^* - M_N^* + M_X^*}, \\[2mm]
\kappa &= \frac{1}{M_R}(M_R^* - M_N^* + M_X^*).
\end{aligned}
\tag{47}
$$

Using these susceptibilities, we obtain the expression for the order parameter for the relative strength of top-down control versus bottom-control,

$$
\frac{D_{eff}^{N,top}}{D_{eff}^{N,top} + D_{eff}^{N,bottom}} = \frac{-\eta_X r_1 \sigma_d^2 \chi}{-\eta_X r_1 \sigma_d^2 \chi + \eta_N \sigma_c^2 \kappa} = \frac{M_X^*}{M_N^*},
\tag{48}
$$

Now for the effective competition, we use the definition

$$
f = \frac{M_R^* + M_X^* - M_N^*}{M_N^* - M_X^*}
\tag{49}
$$

with

$$1/f = \frac{M_N^* - M_X^*}{M_R^* + M_X^* - M_N^*} \quad = \frac{M_R^*}{M_R^* + M_X^* - M_N^*} - 1. \tag{50}$$

to further simplify the susceptibilities,

$$\chi \quad = -\frac{\eta_N \sigma_c^2}{\eta_X \sigma_d^2} \frac{M_X^*}{M_X} \frac{M_N}{M_R} f$$

$$\nu \quad = -\frac{M_R}{\eta_N M_N \sigma_c^2} \frac{1}{f} \tag{51}$$

$$\kappa \quad = \frac{M_R^*}{M_R} \frac{1}{1 + 1/f}.$$

Substituting these expressions into the definitions of effective competition in Eq (5) leads to the expressions in terms of niches in Eq (10).

## 3.2 Numerical details

**3.2.1 ODE simulations.**   In Figs 1(b)–1(c) and 4, we compare the results of simulations and the mean-field equations derived using the cavity method. In order to perform these simulations, we directly numerically integrate the ordinary differential equations in Eq (1). For each trial, we first randomly generate consumer preference random, $d_{aj}$ and $c_jA$, with independent Gaussian distributed elements with mean and variance specified in Eqs (2) and (3). We then numerically solve the system of ordinary differential equations, consisting of $M_X + M_N + M_R$ equations, until steady-state is reached. For both Figs 1(c) and 4, we chose a final time of $t = 10$ with a time step of d$t = 0.1$. We chose the initial values of biomass to be uniformly distributed in the interval [1, 2]. While any positive value will lead to the same steady-state, values closer to steady-state lead to faster convergence. We used the Mathematica function "NDSolve" and solver method "StiffnessSwitching," which works well here because the solution of biomass can be zero or non-zero, leading to very different stiffnesses of the ODEs.

**3.2.2 Cavity equations.**   In every figure, we show some results found by numerically solving our analytic cavity equations. When deriving the cavity equations for the three-level model, we ended up with 9 self-consistent equations, Eqs (25)–(33). Rewriting the susceptibilities in terms of the other variables in Eq (44) results in 6 equations in terms of the variables $\langle X \rangle$, $\langle N \rangle$, $\langle R \rangle$, $\langle X^2 \rangle$, $\langle N \rangle$, and $\langle R \rangle$ which serve as the input into a numerical solver.

Expression our cavity equation in the form $f_i(\vec{x}) = 0$ for $i = 1, \ldots, 6$, we chose to convert the root-finding problem to a constraint optimization problem of the form Minimize$\vec{x} \sum_i [f_i(\vec{x})^2]$, $\vec{x} > 0$ due to better availability of algorithms. We can optionally add the constraint in Eqs (7) and (8) to improve accuracy. We used the default solver method for the Mathematica function "NMinimize," with solver options "AccuracyGoal → 5" and "MaxIterations → 30000" or above. This function requires a range of initial points, which significantly affect the efficiency of the algorithms. While choosing a reasonable range such as [0.1, 1] usually works, one trick we often used to guarantee good initial points is to run a small-scale simulation such as 10 species, which is a rough approximation for the large system scenario that the cavity method assumes. Then we use the value from the simulation for each variable as the upper range and one-half of its value as the lower range.

All the code in this work is written in Mathematica. Demonstrative Mathematica notebooks for both the cavity solution and the ODEs simulations can be found at https://github.com/Emergent-Behaviors-in-Biology/Multi-trophic-ecosystem.

## 4 Discussion

In this paper, we proposed a new model for three-level trophic ecosystems based on generalized Consumer Resource Models. Using the zero-temperature cavity method from spin glass physics, we derived analytic expression for the behavior of this model that are valid for large ecosystem with many species at each trophic level. We found that intra-trophic diversity gives rise to "emergent competition" between species within a trophic level arising from feedbacks mediated by other trophic levels. The strength of this competition depends on both environmental parameters (energy influxes, death rates) and the diversity of the regional species pool. Using analytic solutions, we defined new order parameters for assessing whether an ecosystem is more susceptible to top-down or bottom-up control. Surprisingly, we found that one of these order parameters depends on ecosystem properties only through the fraction of occupied niches. Our analysis suggests that the relative importance of top-down control compared to bottom-up control increases with: (1) higher energy influx into the ecosystem, (2) lower death rate of predators (top level), (3) a larger fraction of species residing in the middle trophic level in the regional species pool, a (4) lower fraction of carnivores and plants in the regional species pool (species in the top and bottom trophic levels). We also found that the amount of top-down control increases as predators in the top trophic level increase their trait diversity, and decreases as herbivores increase their trait diversity.

Our theoretical work can be generalized to accommodate more realistic structures. For instance, our analysis can be generalized to any number of levels, which would allow for investigations into how perturbations propagate through the entire food chain with damping and amplification across levels. Moreover, adding other more complex ecological interactions such as omnivorism, cross-feeding and decomposition could lead to a more realistic and specific understanding of different types of ecosystems [55, 70, 71]. Practically, our theoretical predictions also suggest that a simple way to determine if a three-level system exhibits top-down or bottom-up control is to count the number of carnivores and herbivores. These predictions, summarized in Tables 1 an 2, also provide simple, qualitative rules of thumb for understanding how ecosystem properties change with the shifting species composition of regional species pools and environmental variables.

## Supporting information

**S1 Text. Top-down and bottom-up control in toy model.** In this section, we analyze the special case below where there is only 1 species with intra-species competition on each level. (PDF)

**S1 Fig. Demonstration of bottom-up control and top-down control analogous to Fig 2 for special case with single species on each level. (a)** Exact biomass at each trophic level as a function of $u_1$ with $k_1 = 5$, $m_1 = 1$, $\eta_N = \eta_X = 0.9$, $D_R = 1$, $D_X = 5$, $D_C = 5.c_{11} = 4$, $d_{11} = 4$ (left), and as a function of $k_1$ with $u_1 = 2$ (right). **(b)** Same as (a) except $D_R = 3$, $D_X = 3$, $d_{11} = 9$. (PDF)

**S2 Fig. Plot of effective competition coefficients, (a)** $D_{eff}^R$**, (b)** $D_{eff}^N$ **and (c)** $D_{eff}^X$ **under the same conditions as Fig 4.** (PDF)

**S3 Fig. Plots of effective quantities versus model parameters.** Plot of $D_{eff}^X$, $D_{eff}^N$, $D_{eff}^R$, and $D_{top}^N/(D_{top}^N + D_{bottom}^N)$ versus $r_1 \in [0.4, 1.2]$, $r_2 \in [0.4, 1.2]$, $\sigma_d \in [1, 2.5]$, $\sigma_c$, $u \in [1, 5]$, and $k \in [1, 5]$, obtained from evaluating the cavity solutions with all 6 parameters varied simultaneously, each with 6 possible values in the range. Each trajectory correspond to varying the

parameter on it's x-axis, while fixing the other 5 parameters. These plots arranged in table directly maps to the result in Table 1.
(PDF)

**S4 Fig. System size affects cavity solution convergence.** (a)Histograms of the steady state reached by dynamics of a system with $M_X = 50$ species of carnivores, $M_N = 56$ herbivores and $M_R = 62$ plants, with $k = 4$, $m = 1$, $u = 1$, $\sigma_c = \sigma_d = 0.5$, $\mu_c = \mu_d = 1$, $\eta_X = 1$, $\eta_N = 1$, $\sigma_k = \sigma_m = \sigma_u = 0.1$, and the distribution predicted by our cavity solution. Note that a black dot correspond the finite extinction probability (instead of probability density) predicted by cavity solution, while a black dash correspond to the probability density(b) The average square deviation of the single system statistics from cavity solution as a function of $M_X$ while keeping fixed ratios $r_1 = r_2$ = 0.9, averaged from 200 sample systems.
(PDF)

**S5 Fig. Typical distributions of biomass with uniform distributions of consumer preference matrix elements also agrees well with cavity method.** Solutions shown with parameters $k = 4$, $u = 1$, $\sigma_c = 0.5$, $\mu_c = 5$, $\sigma_d = 0.5$, $\mu_d = 5$, $r_1 = 1$, $r_2 = 1$ and sampled from 200 systems with 30 species in each level.
(PDF)

## Acknowledgments

We thank Maria Yampolskaya, Emmy Blumenthal and Hyunseok Lee for useful discussions. The authors also acknowledge support from the Shared Computing Cluster administered by Boston University Research Computing Services.

## Author Contributions

**Conceptualization:** Zhijie Feng, Robert Marsland, III, Pankaj Mehta.

**Formal analysis:** Zhijie Feng, Robert Marsland, III, Jason W. Rocks.

**Funding acquisition:** Pankaj Mehta.

**Investigation:** Zhijie Feng, Robert Marsland, III, Pankaj Mehta.

**Methodology:** Zhijie Feng, Jason W. Rocks.

**Supervision:** Pankaj Mehta.

**Writing – original draft:** Zhijie Feng, Pankaj Mehta.

**Writing – review & editing:** Zhijie Feng, Jason W. Rocks, Pankaj Mehta.

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
