## [Decision Letter · Decision Letter 0]

21 Jun 2023

Dear Dr. Mehta,

Thank you very much for submitting your manuscript "Emergent competition shapes the ecological properties of multi-trophic ecosystems" for consideration at PLOS Computational Biology. As with all papers reviewed by the journal, your manuscript was reviewed by members of the editorial board and by several independent reviewers. The reviewers appreciated the attention to an important topic. Based on the reviews, we are likely to accept this manuscript for publication, providing that you modify the manuscript according to the review recommendations.

Sincerely,

Jacopo Grilli

Academic Editor

PLOS Computational Biology

Natalia Komarova

Section Editor

PLOS Computational Biology

Reviewer's Responses to Questions

**Comments to the Authors:**

Reviewer #1: One of the outstanding puzzles regarding natural ecosystems is how the interaction of multiple species, each with its own particular attributes, leads collective states of the ecosystem as a whole. One aspect of this puzzle is how interactions among multiple trophic layers, e.g. primary producers, initial consumers, secondary consumers, etc. shape overall ecosystem states. The authors address this question within the well-established framework of consumer resource models (CRMs). What is new is applying to CRMs the perspective that complex interactions can be treated as random parameters. While this approach may be debatable, there is no question that the results of the current study admirably account for a remarkable number of observations about real ecosystems. Furthermore, the authors apply an analytic approach (the cavity method) that both agrees quantitatively with simulations and provides some important simple insights into overall behavior, e.g. whether a particular system is controlled from the bottom up or from the top down. Overall, I believe this work makes an important contribution to our understanding of multi-trophic ecosystems and will be of interest to a broad audience. Finally, the paper was a pleasure to read. I have only a few points for the authors to address:

1. The general reader would benefit from some simplified examples. For example, the coupling between species at the central trophic level due to a shared resource or a shared predator could be explained using a model with only three species, in each case. Even more important would be a simple example of a situation leading to bottom up control vs. one leading to top down control. While ecologists may already understand these ideas, they are likely to be unfamiliar to many readers. Simple "toy models" that capture these behaviors would therefore be a big help. I leave it up to the authors whether some of these examples can be relegated to the supplement.

2. There are a great number of variables, so simplification of the notation would be helpful. For example, in Eq. 1 it would be easier to parse of Q was replaced by P, beta by alpha, and j by i. I don't see any problem with using the same variable name in the summations, and it would be easier to understand the equations. (In fact, the authors have made at least one mistake by having too many variables - in the last box of Fig. 1e the Q should be a P. Above Eq. 2, the two appearances of d_ia should be d_alpha i).

3. The histograms in Fig. 1c appear to be the combined results of many independent simulations (and this number should be stated in the caption). However, a given ecosystem only has one set of species. What does that distribution look like for one realization of parameters? It would be helpful to show such individual distributions - ideally along with some measure of agreement between individual distributions and the expected average distribution - as a function of the total number of species. Also, in Fig. 1c, there is a difference between the simulation results and the cavity results for X and N at the lowest bar in the histogram. Given the essentially perfect agreement for the rest of the distribution, some discussion of this deviation is called for.

4. There is a great deal of discussion in the ecology literature about rank-abundance curves, so it would be appropriate to discuss how the distributions of species abundances in the current model compare to actual rank-abundance curves.

5. There are a few misspellings and missing words, so another careful reading of the manuscript would be helpful.

Reviewer #2: The review is uploaded as an attachment.

Reviewer #3: In this manuscript, Feng et al examine the role of trophic structure on ecosystem assembly. The authors' analysis exploits the cavity method, which helps analyzing ecosystems with randomized interactions, and provides powerful intuition about how different observables (e.g., species diversity, biomass) depend on the underlying parameters (e.g., mean and variance in predation rate, species pool size). In particular, the authors use this technique to quantify the relative extent to which an ecosystem is sensitive to changes in primary producers (bottom-up) versus predators or tertiary consumers (top-down). Their analysis shows that the key quantities that determine ecosystem sensitivity or control are the species packing fractions, i.e., the fraction of occupied of niches in each trophic level.

The manuscript is clearly-written and addresses a long-recognized ecological question. I found the "order parameters" suggested by the authors particularly useful and simple to quantify, and the cavity method provides a solid theoretical basis to deploy them. The work also synthesizes and unifies a lot of observations and hypotheses about trophic levels in the ecological literature through a nice extension of MacArthur's classical model, and is thus well-suited for publication in PLoS Computational Biology.

However, to be more widely appreciated by ecologists, I suggest a few points of clarifications about assumptions, as well as a few outstanding questions that the authors could either choose to address in a revision, or alternatively leave a comment in the Discussion. The paper already contains enough results to make the manuscript suitable for publication --- my comments contain suggestions and curiosities.

Major comments

(1) Energy conservation and the 10% rule: Does the model conserve energy? It seems to me that it does not, in the sense that there is no strict bound on how much biomass can be derived from one trophic level and transferred to the next. This seems a bit strange, since in ecology, people typically think that every subsequent trophic level has ~10% the biomass of the previous one, as a result of dissipation and imperfect energy conversion. It would be nice to explore the consequences of such energy loss, since it would make the biomasses of the three trophic levels systematically different. I am curious to see to what extent this would influence the Dtop/Dbottom metrics. In case the authors do not wish to explore this, they could add a note somewhere in the paper.

(1.5) Consumption-conversion asymmetry: On a similar note, the authors assume no asymmetry between consumption and growth (biomass conversion), i.e., there is one set of c's and d's. To what extent does that influence the results? It would be nice to add a note about this assumption as well.

(2) Keystone species: Ecosystems can often experience dramatic changes due to the removal or addition of one "keystone" species. I am curious to understand if any of the surviving species in the simulated ecosystems in this paper are of this kind. Specifically, does the theory expect many keystone species or few? Due to the species in a trophic level being statistically similar, it was not clear to me. It seems like in addition to top down and bottom up control, where the carrying capacity or death rate of all species in a trophic level change, one could also compare the scale of these changes with keystone control, where only the presence of one species changes.

(3) Relation between sensitivity and number of levels: The authors show that many of their steady state properties are robust to including more trophic levels. I, however, could not understand if including more levels would suppress the sensitivity to the bottom-up or top-down controls considered here. For instance, would it be the case that whether an intermediate trophic level is controlled by top or bottom depends not just on the species packing fractions of the top or bottom levels, but also on the trophic distance to these levels?

Minor comments

(1) Is it possible to get multiple stable states in the model? The authors state that their model only has a unique stable steady state, but I saw no reference to a proof in the appendix, or a citation to the claim.

(2) It seems by looking at the figures that top-down control is rare --- most of the phase diagrams show blue (bottom-up control). Is this just by chance due to the parameters sampled, or is this a broader statement? Even in Fig. 3 the order parameters span three orders on the left of 1, while 2 orders to its right.

(3) In Fig. 2c and d, I would suggest plotting the yellow curves (middle trophic level) separately from all the others, since that is the one the reader should focus on. Since all the curves are changing, it took me a while to understand that the yellow one was the focus. This could be done by including the other curves in an inset, or just separately.

**Have the authors made all data and (if applicable) computational code underlying the findings in their manuscript fully available?**

Reviewer #1: Yes

Reviewer #2: Yes

Reviewer #3: Yes

PLOS authors have the option to publish the peer review history of their article (what does this mean?). If published, this will include your full peer review and any attached files.

Reviewer #1: No

Reviewer #2: **Yes: **Onofrio Mazzarisi

Reviewer #3: No

Figure Files:

Data Requirements:

Reproducibility:

References:

---

## [Decision Letter · Decision Letter 1]

10 Nov 2023

Dear Dr. Mehta,

We are pleased to inform you that your manuscript 'Emergent competition shapes top-down versus bottom-up control in multi-trophic ecosystems' has been provisionally accepted for publication in PLOS Computational Biology.

Best regards,

Jacopo Grilli

Academic Editor

PLOS Computational Biology

Natalia Komarova

Section Editor

PLOS Computational Biology

Reviewer's Responses to Questions

**Comments to the Authors:**

Reviewer #1: The authors have satisfactorily addressed all my comments, and I am happy to recommend publication.

Reviewer #2: The Authors nicely addressed my comments.

Reviewer #3: I thank the authors for suitably addressing the comments by me and other reviewers. I believe the paper is now greatly strengthened and suitable for publication in PLOS Computational Biology. Congratulations to the authors for this nice work.

**Have the authors made all data and (if applicable) computational code underlying the findings in their manuscript fully available?**

Reviewer #1: Yes

Reviewer #2: Yes

Reviewer #3: Yes

PLOS authors have the option to publish the peer review history of their article (what does this mean?). If published, this will include your full peer review and any attached files.

Reviewer #1: No

Reviewer #2: **Yes: **Onofrio Mazzarisi

Reviewer #3: No

---

## [Editor Report · Acceptance letter]

8 Jan 2024

PCOMPBIOL-D-23-00463R1 

Emergent competition shapes top-down versus bottom-up control in multi-trophic ecosystems

Dear Dr Mehta,

I am pleased to inform you that your manuscript has been formally accepted for publication in PLOS Computational Biology. Your manuscript is now with our production department and you will be notified of the publication date in due course.

With kind regards,

Bernadett Koltai
